

# A mixed methods study of multiple health behaviors among individuals with stroke

Matthew Plow[1], Shirley M. Moore[1], Martha Sajatovic[2] and Irene Katzan[3]

[1] School of Nursing, Case Western Reserve University, Cleveland, OH, United States of America
[2] Department of Psychiatry and of Neurology, Neurological and Behavioral Outcomes Center, Case Western Reserve University, Cleveland, OH, United States of America
[3] Neurological Institute, Center for Outcomes Research and Evaluation, Cleveland Clinic, Cleveland, OH, United States of America

Corresponding author
Matthew Plow, map208@case.edu

## ABSTRACT

**Background**. Individuals with stroke often have multiple cardiovascular risk factors that necessitate promoting engagement in multiple health behaviors. However, observational studies of individuals with stroke have typically focused on promoting a single health behavior. Thus, there is a poor understanding of linkages between healthy behaviors and the circumstances in which factors, such as stroke impairments, may influence a single or multiple health behaviors.

**Methods**. We conducted a mixed methods convergent parallel study of 25 individuals with stroke to examine the relationships between stroke impairments and physical activity, sleep, and nutrition. Our goal was to gain further insight into possible strategies to promote multiple health behaviors among individuals with stroke. This study focused on physical activity, sleep, and nutrition because of their importance in achieving energy balance, maintaining a healthy weight, and reducing cardiovascular risks. Qualitative and quantitative data were collected concurrently, with the former being prioritized over the latter. Qualitative data was prioritized in order to develop a conceptual model of engagement in multiple health behaviors among individuals with stroke. Qualitative and quantitative data were analyzed independently and then were integrated during the inference stage to develop meta-inferences. The 25 individuals with stroke completed closed-ended questionnaires on healthy behaviors and physical function. They also participated in face-to-face focus groups and one-to-one phone interviews.

**Results**. We found statistically significant and moderate correlations between hand function and healthy eating habits ($r = 0.45$), sleep disturbances and limitations in activities of daily living ($r = -0.55$), BMI and limitations in activities of daily living ($r = -0.49$), physical activity and limitations in activities of daily living ($r = 0.41$), mobility impairments and BMI ($r = -0.41$), sleep disturbances and physical activity ($r = -0.48$), sleep disturbances and BMI ($r = 0.48$), and physical activity and BMI ($r = -0.45$). We identified five qualitative themes: (1) Impairments: reduced autonomy, (2) Environmental forces: caregivers and information, (3) Re-evaluation: priorities and attributions, (4) Resiliency: finding motivation and solutions, and (5) Negative affectivity: stress and self-consciousness. Three meta-inferences and a conceptual model described circumstances in which factors could influence single or multiple health behaviors.

**Discussion**. This is the first mixed methods study of individuals with stroke to elaborate on relationships between multiple health behaviors, BMI, and physical function. A

conceptual model illustrates addressing sleep disturbances, activity limitations, self-image, and emotions to promote multiple health behaviors. We discuss the relevance of the meta-inferences in designing multiple behavior change interventions for individuals with stroke.

## INTRODUCTION

Approximately 795,000 people in the United States have a stroke each year (*Mozaffarian et al., 2015*). Individuals with stroke are at a significant risk for experiencing a secondary stroke and often have multiple cardiovascular risks factors, such as obesity, hypertension, sleep apnea, and dyslipidemia (*Kernan et al., 2014*). Reducing these risks requires engagement in multiple health behaviors (*Hackam & Spence, 2007*). Accordingly, several studies have evaluated behavior change interventions to reduce cardiovascular risk factors in individual with stroke. However, these interventions have largely been ineffective or inconclusive in changing multiple behaviors and reducing cardiovascular risks (*Lager et al., 2014*). Clearly, fundamental questions remain regarding the best ways to promote multiple health behaviors that reduce risks. For example, it is unknown whether it is more effective to reduce risks by targeting change in multiple behaviors sequentially or simultaneously.

Observational studies of individuals with stroke have typically focused on a single health behavior (*Lager et al., 2014*). Quantitative studies have identified factors associated with physical activity (*Field et al., 2013*), medication adherence (*O'Carroll et al., 2014*), and caregiver interactions (*Mackenzie & Greenwood, 2012*). Similarly, qualitative studies have described the facilitators for and barriers to engaging in a single health behavior (*Chambers et al., 2011*; *Damush et al., 2007*). These studies have documented the negative influence of stroke impairments on behavioral outcomes, such as exercise (*Damush et al., 2007*; *Field et al., 2013*) and medication adherence (*Chambers et al., 2011*; *O'Carroll et al., 2014*).

However, few studies have addressed questions pertinent to promoting multiple behaviors. For example, rarely are there attempts to understand the linkages between healthy behaviors or how perceptions about one behavior may shape perceptions about another behavior. To the best of our knowledge, no studies have examined the circumstances in which stroke impairments may impede the engagement in single or multiple health behaviors. Furthermore, no studies of individuals with stroke have focused on promoting healthy eating and sleeping habits, two behaviors that may be interrelated with physical activity to reduce cardiovascular risks in individuals with stroke (*Grandner et al., 2011*; *Mozaffarian et al., 2011*). Understanding the linkages between stroke impairments and healthy behaviors, such as physical activity, sleep, and nutrition, will be important in reducing cardiovascular risks among strong survivors (*Grandner et al., 2011*; *Mozaffarian et al., 2011*).

In order to understand the complexities of engaging in multiple health behaviors while coping with stroke impairments, we conducted a mixed methods study of 25 individuals

with stroke. A mixed methods design or the collection and integration of qualitative and quantitative data is particularly relevant to understanding how to promote multiple health behaviors because of the need to determine both the extent and the circumstances in which behaviors are interrelated. Specifically, we implemented a convergent parallel design in which the same 25 individuals with stroke completed closed-ended questionnaires and participated in face-to-face focus groups and one-to-one phone interviews. We decided to focus on physical activity, sleep, and nutrition because of their importance in achieving energy balance, maintaining a healthy weight, and reducing cardiovascular risks (*Grandner et al., 2011*; *Mozaffarian et al., 2011*).

The three purposes of this study are as follows: (1) to examine the relationships between stroke impairments and physical activity, sleep, and nutrition; (2) to examine the relationships between physical activity, sleep, and nutrition; (3) to gain further insight into possible strategies to promote multiple health behaviors among individuals with stroke. Each purpose was accomplished by employing both qualitative and quantitative methods that were analyzed independently and then were integrated during the inference stage to identify meta-inferences. The meta-interferences were developed by examining areas of convergence and divergence in the qualitative and quantitative results, with guidance from *Noar, Chabot & Zimmerman*'s (*2008*) multiple behavior change framework.

*Noar, Chabot & Zimmerman (2008)* described three approaches to examine multiple health behaviors: behavioral change principles, global health/behavioral categories, and multiple health behaviors. In the behavioral change principles approach, the focus is on examining whether there is a common set of behavioral change principles that can be applied to promote multiple behavioral changes. In the global health/behavioral category approach, the focus is on examining whether there are global health cognitions that predict attitudes towards behavioral categories (e.g., physical activity) and specific behaviors (e.g., walking), which in turn predict the actual engagement in healthy behaviors. In the multiple behavioral approach, the focus is on examining the linkages between behaviors. We used all three of *Noar, Chabot & Zimmerman*'s (*2008*) approaches to obtain a comprehensive understanding of engaging in multiple health behaviors.

## METHOD

### Overview

A mixed methods convergent parallel fixed design was implemented in which qualitative data was prioritized over quantitative data in order to develop a conceptual model. We aimed to collect different but complementary data to facilitate triangulation and enrich the interpretation of results. Qualitative data was prioritized because of the limited amount of research on multiple health behaviors and the need to develop new hypotheses on how to promote multiple health behaviors. Qualitative and quantitative data were collected in the same order for all participants. All 25 participants first completed the close-ended questionnaires and then participated in one of three face-to-face focus groups, followed by a one-to-one phone interview. Thus, each participant completed the questionnaires, attended a focus group, and received a follow up phone call. The questionnaires were administered to
measure healthy behaviors and physical function. The focus groups were conducted to obtain narratives about engaging in multiple health behaviors. The one-to-one interviews were conducted to help validate interim analyses of the focus groups and to elaborate on the emerging themes. The qualitative data (i.e., focus group and phone interview) and quantitative data (i.e., questionnaires) were analyzed independently and then integrated during the interpretation phase to develop meta-interferences by using *Noar, Chabot & Zimmerman*'s (*2008*) framework. An Institutional Review Board at Cleveland Clinic approved the research protocol (IRB # 11-847). All participants provided written informed consent.

## Participants

The study criteria were as follows: a self-report diagnosis of stroke, the ability to communicate over the phone, and a minimum of 18 years of age. We used several strategies to recruit participants for the study. The participants in the first focus group were recruited from a physical therapist-led group exercise class designed for individuals with stroke. The participants in the second and third focus groups were recruited via flyers posted in the waiting rooms of outpatient neurological and rehabilitation healthcare services.

## Questionnaires

Physical activity was measured using the Godin Leisure-Time Exercise Questionnaire (*Godin & Shephard, 1985*). Three questions were asked about the frequency of engaging in strenuous (e.g., running and vigorous swimming), moderate (e.g., fast walking, and tennis), and light (e.g., fishing and slow walking) leisure-time exercises for at least 15 min. The strenuous, moderate, and light activities were multiplied by 9, 5, and 3, respectively, and then summed to provide a composite score. A higher score indicated a greater level of engagement in physical activity. The validity and reliability of the Godin Leisure-Time Exercise Questionnaire was validated in several studies and tested in diverse population groups, including those with neurological disabilities. For example, the questionnaire was validated against accelerometers ($r = 0.53$) and showed adequate test-retest reliability ($r = 0.77$) (*Godin, 2011*; *Gosney et al., 2007*; *Noreau et al., 1993*; *Plow, Finlayson & Cho, 2012*).

Sleep disturbance was measured using the Neuro-QOL eight-item short form, version 1 (*Cella et al., 2012*). The questions pertained to the past seven days and were about having trouble getting up in the morning, having trouble stopping thoughts at bedtime, experiencing sleepiness during the day, having trouble falling asleep, and having pain that prevents sleep. The responses to the items ranged from never (1) to always (5) and were summed; higher scores indicated greater problems with sleeping. Neuro-QOL is a set of self-reported, health-related quality of life measures that have been validated (i.e., $\alpha = 0.78$; test–retest reliability ICC = 0.61) in individuals with stroke (*Neuro-QoL, 2015*; *Cella et al., 2012*).

Healthy nutrition habits were measured using a five-item nutritional survey. The questions asked about the frequency of making healthy food choices, eating five servings of fruits and vegetables a day, limiting fat intake, reading labels, and eating regularly (*Nosek et al., 2006*). The responses to the items were on a scale ranging from never (0), sometimes (1), and often (2) and were summed; higher scores indicated more frequent engagement in healthy nutrition habits. The survey was previously tested in a large study of adults with

disabilities (*Nosek et al., 2006*) and adults with multiple sclerosis (*Plow, Finlayson & Cho, 2012*). Its reliability ($\alpha = 0.74$; test–retest reliability $r = 0.77$) was found to be adequate.

Physical function was measured using the 16-item Stroke Impact Scale (*Duncan et al., 2003*). The questions pertained to the past two weeks and asked about difficulties in bathing, shopping, walking one block, getting in and out of a car, climbing one flight of stairs, and controlling bladder and bowels. The 16 items were summed and divided into three subscales–activities of daily living, mobility, and hand function–ranging from 0 to 100; higher scores indicated less pronounced effects of stroke impairments on daily activities. A Rasch analysis was used to construct and validate the 16-item scale based on the physical domain composite of the Stroke Impact Scale. Concurrent validity (i.e., significantly different across disability levels) and reliability were found to be adequate (*Duncan et al., 2003*; *Edwards & O'Connell, 2003*).

## Questionnaire analysis

All data collected from the responses to the questionnaire met the assumption of normality. Thus, Pearson $R$ correlations were used to examine the associations between the responses to the questionnaires and self-reported body mass index (BMI). Statistically significant correlations were shown as two-tailed $p$-values. A Pearson $R$ value less than 0.3 was considered small, a value between 0.3 and 0.5 was considered moderate, and a value greater than 0.5 was considered large (*Cohen, 1988*).

## Qualitative interview procedure

All interviews were semi-structured (i.e., open-ended questions followed by probes and transitions), audio-recorded, and transcribed verbatim. The focus groups lasted approximately two hours, which were conducted by the first author while a research assistant took notes. The one-to-one interviews lasted approximately 45 min, which were conducted over the phone by a trained research assistant.

### Focus groups

We first asked open-ended questions about the participants' habits and motivators for and barriers to engaging in physical activity, nutrition, and sleep. For example, we asked participants whether engaging in physical activity was important and a priority, which types of physical activity they preferred engaging in, and what motivated or hindered engagement in physical activity. Probes included asking about the use of resources to learn about and engage in all three behaviors. We then asked participants whether and how they perceived physical activity, nutrition, and sleep to be interrelated and encouraged them to provide examples from their own lives. After each section, the interviewer summarized areas of consensus and disagreement and asked for further input, which helped identify new themes and refine the interview guides. For example, in the second and third focus groups, questions were revised to elaborate on possible themes about spirituality, autonomy, self-image, strategies to manage emotions, and how impairments could hinder multiple behaviors.

### One-to-one phone interviews

We first summarized the findings from the focus group and asked participants if they agreed with the summary and wanted to add anything. Participants were then asked about their social, leisure, occupation, and domestic life roles to establish a rapport with them and to avoid asking irrelevant questions. We then transitioned to a discussion about healthy behaviors by asking the participants to define healthy behaviors and to provide specific examples. Probes were asked to explore global attitudes and perceptions about engagement in healthy behaviors and the perceived relationship of healthy behaviors with recovery and the ability to cope with stroke impairments. Questions were then asked about physical activity, nutrition, and sleep. Compared to the focus groups, more specific questions were asked during the one-to-one interviews to elicit participants' attitudes, knowledge, confidence, outcome expectations, problem-solving strategies, and perceived barriers and facilitators in their social and physical environments that affected engagement in each of the three behaviors.

## Thematic qualitative analysis

The thematic analysis was based on recommendations by *Elo & Kyngas (2008)*, while ensuring the trustworthiness of the thematic analysis was based on recommendations by *Shenton (2004)*. The analysis consisted of an inductive-category and theme-development approach to develop the conceptual model. Focus groups and phone interview transcripts were first read multiple times to obtain an overall sense of the data. During the initial reading, the first author and the research assistant performed open coding; i.e., notes were written in the margins of the text. In subsequent readings, similarities and differences between focus group and phone interviews were also noted. Interim analyses identifying similarities and differences between the focus groups and phone interviews helped refine the interview guide and determine when data saturation had occurred.

Notes were compared and discussed to generate descriptive labels that encompassed the data from the focus groups and one-to-one interviews. Overall patterns among the descriptive labels were used to identify clusters to organize the data into themes. Each theme was then operationally defined to facilitate consistent coding of the data. Transcripts were coded using Atlas.ti (Version 7; Scientific Software Development GmbH, Berlin, Germany), which helped generate an audit trail, facilitate the conceptual mapping of categories, and provide flexibility in revising the coding scheme as analyses and discussions occurred.

After the focus group and one-to-one interview transcripts were coded, the first author and the research assistant discussed disagreements and the development of subthemes. Sections of data that were not coded were reviewed, and when necessary, themes and subthemes were revised to provide a comprehensive description of the data. Once the themes and subthemes were refined and finalized, transcripts and quotes within each theme were re-read to develop the conceptual model. Co-authors who were not involved in the coding of the data examined the appropriateness of each theme and the conceptual model by reviewing exemplar quotes and drawing upon their expertise in nursing, neurology, and psychiatry to help establish content validity.

Several steps were taken to help ensure the trustworthiness of the qualitative analysis. Data collection and analysis proceeded iteratively to determine data saturation and

identify themes; focus groups were followed by one-to-one interviews (i.e., member checks); an audit trail was generated (i.e., conformability); transcripts were re-read as a whole, and quotes within categories were reviewed and scrutinized on multiple occasions (i.e., dependability); there were frequent debriefing sessions, and disagreements were discussed until consensus was reached (i.e., peer scrutiny); and sections of data that were not coded for the possible inclusion of a new theme (i.e., negative case analysis) were examined.

### Integrating qualitative and quantitative results

At the inference stage, we explored areas of convergence and divergence in the findings of the qualitative and quantitative analyses to develop and refine meta-inferences and the conceptual model. Data transformation and typology development were used to integrate the results (*Caracelli & Greene, 1993*). We first qualitized the quantitative results by writing narrative summaries that described the characteristics of each participant derived from the responses to the questionnaires. We then compared the narrative summaries of participants and generated possible explanations for the similarities and differences in the patterns for engaging in multiple health behaviors. We further refined the explanations using the calculated means and correlations and then selected the most logical explanations.

Qualitizing explanations were then organized in a side-by-side comparison table with qualitative themes and exemplar quotes using *Noar, Chabot & Zimmerman*'s (*2008*) framework. Specifically, we organized the qualitizing explanations, qualitative themes, and exemplar quotes based on their relevance in illustrating *Noar, Chabot & Zimmerman*'s (*2008*) three approaches for understanding multiple health behaviors. We then looked for patterns within each of the approaches to develop the conceptual model and meta-inferences. The conceptual model and meta-inferences were refined using a negative case analysis approach; i.e., we searched for and corrected for any contradictions that were found between the conceptual model, the meta-inferences, and the narrative summaries of each participant. Thus, integrating results was an iterative process of comparing and contrasting the qualitative and quantitative data using a theoretical framework.

Legitimation strategies where used in integrating the data (*Onwuegbuzie & Johnson, 2006*). These strategies included confirming the accuracy of the narrative summaries by a multidisciplinary team of experts and individuals with stroke (i.e., inside-outside legitimation) by using the qualitative results to elaborate on the quantitative associations (i.e., weakness minimization legitimation); analyze the qualitative and quantitative data separately and then develop meta-inferences (i.e., paradigmatic mixing legitimation); use validated questionnaires and strategies to enhance the trustworthiness of the qualitative results (i.e., multiple validities legitimation); and draw upon the existing theoretical literature on multiple health behaviors (i.e., commensurability legitimation).

## RESULTS

### Quantitative results

The socio-demographic characteristics of mobility device use, and BMI (self-reported) of the 25 participants are reported in Table 1. The mean age of the research sample was 64 years; the ratio of men to women was approximately equal. There was a substantial

**Table 1    Characteristics of the participants (n = 25).**

|  | Mean (range) |
| --- | --- |
| *Age (years)* | 64.12 (46–89) |
| *Years since first stroke* | 5.68 (1–33) |

|  | N Frequency count (%) |
| --- | --- |
| *Gender* |  |
| Female | 12 (48) |
| *Racial minority* | 5 (20) |
| *Education (>15 years)* | 8 (57) |
| *Living with someone else* | 19 (76) |
| *Living with spouse* | 16 (64) |
| *Body mass index* |  |
| Underweight | 1 (4) |
| Normal | 6 (24) |
| Overweight | 10 (40) |
| Obese | 8 (32) |
| *Cane use*[a] |  |
| Never | 11 |
| Sometimes | 5 |
| Always | 7 |
| *Walker use* |  |
| Never | 14 |
| Sometimes | 6 |
| Always | 4 |
| *Wheelchair use* |  |
| Never | 17 |
| Sometimes | 5 |
| Always | 2 |

**Notes.**

[a] Participant could report using multiple mobility aids and may have skipped questions.

amount of variation in the number of years since having a stroke (range 1–33 years). Almost two-thirds of the participants were either overweight or obese. Most participants engaged in low amounts of physical activity but indicated that they engaged in some healthy eating habits sometimes or routinely. Sleep disturbances were a common problem. The responses to questions about impairment and healthy behavior are summarized in Table 2, and the Pearson R correlations are reported in Table 3.

Regarding the first purpose of the study about examining the relationships between stroke impairments and healthy behaviors, we found statistically significant and moderate correlations between hand function and healthy eating habits ($r = 0.45$), sleep disturbances and limitations in activities of daily living ($r = -0.55$), BMI and limitations in activities of daily living ($r = -0.49$), physical activity and limitations in activities of daily living ($r = 0.41$), and mobility impairments and BMI ($r = -0.41$). Regarding the second purpose of the study about examining the relationships between physical activity, sleep, and nutrition, we

**Table 2   Mean, standard deviation, and range of healthy behaviors and physical function questionnaires.**

| Variable | M (SD) | Range |
| --- | --- | --- |
| Healthy eating habits | 7.96 (1.49) | 5.00–10.00 |
| Physical activity | 33.20 (27.54) | 0.00–100.00 |
| Sleep disturbances | 15.20 (5.26) | 8.00–27.00 |
| ADL limitations | 78.02 (16.37) | 37.50–100.00 |
| Mobility | 72.40 (19.09) | 39.29–100.00 |
| Hand function | 49.00 (35.71) | 0.00–100 |

Notes.

Sleep disturbance, neuro-QOL eight-item short form; Physical activity, Godin Leisure-Time Exercise Questionnaire; Healthy eating habits, five-item questionnaire on healthy eating habits; ADL limitations, Activities of Daily Living subscale of Stroke Impact Scale-16; Mobility, subscale of Stroke Impact Scale-16; Hand function, subscale of Stroke Impact Scale-16.

**Table 3   Pearson correlations between healthy behaviors and health-related quality of life questionnaires.**

| Variables | BMI | Physical activity | Sleep | Healthy eating habits | ADL limitations | Mobility |
| --- | --- | --- | --- | --- | --- | --- |
| Physical activity | −0.45[*] | | | | | |
| Sleep disturbances | 0.48[*] | −0.48[*] | | | | |
| Healthy eating habits | −0.10 | 0.33 | −0.16 | | | |
| ADL limitations | −0.49[*] | 0.41[*] | −0.55[**] | 0.42[*] | | |
| Mobility | −0.41[*] | 0.39 | −0.19 | 0.33 | 0.78[**] | |
| Hand function | 0.05 | 0.20 | −0.20 | 0.45[*] | 0.70[**] | 0.71[**] |

Notes.

[*]Correlation is significant at the 0.05 level (two-tailed).

[**]Correlation is significant at the 0.01 level (two-tailed).

BMI, body mass index (self-report); Sleep disturbance, neuro-QOL eight-item short form; Physical activity, Godin Leisure-Time Exercise Questionnaire; Healthy eating habits, five-item questionnaire on nutrition-related behaviors; ADL limitations, Activities of Daily Living subscale of Stroke Impact Scale-16; Mobility, subscale of Stroke Impact Scale-16; Hand function, subscale of Stroke Impact Scale-16.

found statistically significant and moderate correlations between sleep disturbances and physical activity ($r = -0.48$), sleep disturbances and BMI ($r = 0.48$), and physical activity and BMI ($r = -0.45$). We found non-significant and/or small correlations between physical activity and healthy eating habits ($r = 0.33$) and sleep disturbances and healthy eating habits ($r = -0.16$).

## Qualitative results
### Overview
Engaging in multiple health behaviors was a dynamic trial and error process in which participants would set goals to engage in healthy behaviors and would re-assess and change goals as needed. This trial and error process was driven by a desire to limit the negative impact of impairments while at the same time being presented with profound barriers that limited the desire and ability to engage in multiple health behaviors. Participants often expressed frustration at being unable or having to put in extra effort to engage in activities as desired. For some participants, this led to greater motivation to engage in multiple health behaviors, whereas for other participants it led to greater amounts of stress. Although some participants believed that engaging in multiple health behaviors was completely under their control, several participants recognized that engaging in multiple health behaviors

was influenced by multiple environmental factors that were not always under their control. Healthcare providers, friends and family were described as facilitating and hindering engagement in multiple health behaviors. Most participants spent considerable time seeking information about ways to reduce the negative impact of their stroke.

We identified the five following overarching themes, with each theme having two to four subthemes: (1) Impairments: reduced autonomy, (2) Environmental forces: caregivers and information, (3) Re-evaluation: priorities and attributions, (4) Resiliency: finding motivation and solutions, and (5) Negative affectivity: stress and self-consciousness. Themes one, two, and three help address the first study purpose to examine the relationships between stroke impairments and healthy behaviors, while categories four and five help address the second study purpose to examine the relationships among healthy behaviors. Table 4 summarizes the categories and subcategories using exemplar quotes.

(1) Impairments: reduced autonomy.

Subcategory 1: limiting options. Fatigue, mobility impairments, and pain were often described as limiting options, abilities, and confidence to engage in physical activity, nutrition, and sleep. Participants described circumstances in which impairments hindered single or multiple health behaviors. For example, sometimes fatigue was described as only limiting engagement in physical activity, while other times fatigue was described as limiting both physical activity and the ability to eat healthily. Similarly, some participants described how mobility impairments prevented engaging in desired modes of physical activity, such as walking or swimming, due to safety concerns (e.g., fear of falling or drowning). Other participants described how mobility impairments and fatigue made it difficult to cook and/or go grocery shopping, which increased the likelihood of unhealthy food choices (e.g., eating fast food or highly processed food). Pain was described as preventing adequate sleep, engagement in physical activity, and/or the desire to eat healthy foods.

Subcategory 2: changes in social roles. Domestic, leisure, and occupational roles were altered in desired and undesired ways after the stroke, which facilitated or hindered engagement in multiple health behaviors. For some participants, being unable to work or to accomplish daily tasks and chores resulted in more free time to engage in physical activity and/or they were less tempted to eat unhealthy foods. Participant #3 said,

"I think it's actually easier now that I'm not working to eat healthy. I think work was more of a stressor. I made the wrong choice a little more. I think my friends have less influence than they did because I am just not exposed to those activities, like going to the bar with the guys or having a hot dog after golfing".

Alternatively, others noted being unable to engage in leisure activities, such as golf and gardening, and having fewer daily tasks to accomplish, such as cleaning the house, which made them more sedentary. Some participants viewed limitations in daily activities and a more sedentary lifestyle as a reason for sleeping difficulties.

(2) Environmental forces.

Subcategory 1: formal caregivers. Most participants received rehabilitation services and reported occasionally engaging in the prescribed home exercise program. However, participants rarely reported seeing a dietician or a sleep specialist and infrequently had conversations with their physician about engaging in healthy behaviors. Physicians were

**Table 4  Summary of themes and subthemes.**

| Themes | Subthemes | Examples |
|---|---|---|
| Impairments: reduced autonomy | Limiting options | "I'm just too slow at getting everything else done, and I think I just kind of cut back on everything that I want to do. The hobbies, exercises, and socializing get cut first". |
| | Changes in social roles | "Before my stroke, I was a real estate sales lady. I had my own car, I could drive. After the stroke, I had to give up my car, give up driving, give up my job. Now I hardly do anything". |
| | | "I think it's actually easier now that I'm not working to eat healthy. I think work was more of a stressor. I made the wrong choice a little more. I think my friends have less influence than they did". |
| Environmental Forces | Formal caregivers | "And the doctor said, if you change your diet, go vegan, and eliminate oils, you would probably be able to get off of all the medication. So I converted to veganism in the hospital". |
| | | "My doctor told me that no matter what I did, I'd be on medication for all my life. And I refused to accept that and take the medications. As long as I do the exercises and I eat right, my blood pressure will go down". |
| | Informal caregivers | "I started coming to this exercise class. Sometimes I don't feel like coming but my daughters drag me, and then I have to come. I feel 100% better". |
| | | "Well, my kids, they always seem a little surprised at whatever I do […] They sometimes do too much for me". |
| | Seeking information | "I think I've learned more about health since I've had my stroke than ever before in my life because I think that you can't beat it unless you understand it". |
| Re-evaluation: priorities and attributions | Priorities and standards | My eating habits have substantially changed for the worse. And it's whatever is most convenient, easy for me to do. […]. Basically, if it's anything more than a TV dinner, I am at a complete loss". |
| | | "I don't mean to sound like a snob, but my standards are much higher now. I used to think it was okay to eat greasy, fried foods, and now I wouldn't touch it […]". |
| | Stroke cause | "I don't know exactly all the problems that cause a stroke, but I know that my sugar and sodium intake contributed to it. […] So I had to start learning how to eat properly, and that's my goal". |
| | | "According to what I've read, the lack of exercise sometimes is a factor of diabetes, and it's also a factor of having a stroke. So if I want to avoid that pitfall in the future, it's incumbent upon me to keep myself in the best physical condition that I can […]". |
| | Self-image | "So establishing that track record of doing it, of constantly completing what you set out to do, you start feeling like you are making some progress, you feel more confident, and you feel better about yourself". |
| Resiliency: finding motivation and solutions | Self-determination | "Well, the thing in which–that keeps me on track is my desire to get better. I want to get back to doing what I used to do. And, I believe I can do it, but I just have to be cautious and persistent about doing it". |
| | | "I just refuse to accept restrictions, and I've been that way my whole life. I've never accepted no". |
| | Environmental planning | "I only take nutritious food into the house now". |
| | | "I just like to make sure that there's fruit and vegetables in the house and that I have access to them. And I–I've been trying to eat those rather than eating junk food, which I've been trying not to keep in the house". |
| | Outcome expectations | "I'm not exactly where I want to be–everything has not returned to my left side yet, but exercising and eating right and watching my weight is getting me closer". |
| | | "Getting enough sleep and exercising are the things that contribute to me being able to mentally be stronger and help resist stress or negativity". |
**Table 4** (*continued*)

| Themes | Subthemes | Examples |
|---|---|---|
| *Negative Affectivity* | Stress and anxiety | "About the first three, man, it was fear. I couldn't sleep. After having the stroke, I was so worried about having another one". |
| | | "I feel almost like a prisoner in my own house. And when you can't do something and you want to do something, it makes you frustrated. […] And not being able to get out and about and do additional things like exercise and go for a walk makes me more frustrated". |
| | Self-consciousness | "I just don't like going to the gym. I don't like it when other people notice me struggling. They take pity on me, and I don't like that". |
| | | "It's just this and the effect the stroke had on me, and it's made me very sensitive [about my body]. You know, I find myself questioning myself on everything I do–whether I'm doing the right thing or whether I'm doing it well enough". |

perceived as being too busy or not knowledgeable in nutrition and physical activity topics, and participants were concerned about discussing sleep problems because of not wanting to be prescribed additional medications. Nonetheless, when a healthcare professional provided advice, participants sometimes described it as a "nudge" to seek further information and/or initiate multiple behavior changes. For example, several participants described trying new diets (e.g., vegetarian) and engaging in different types of physical activity (e.g., yoga and swimming) because their physician recommended it.

Alternatively, some participants described engagement in multiple health behaviors as an act of defiance against healthcare advice and recommendations. For example, some participants described being more motivated to engage in physical activity after being told they probably would not see functional improvements after six months. Other participants refused to take prescribed medication and instead ate healthier and engaged in more physical activity. Participant #6 said,

"My doctor told me that no matter what I did, I'd be on medication for all my life. And I refused to accept that and take the medications. As long as I do the exercises and I eat right, my blood pressure will go down".

Medications were also described as influencing sleep quality and nutrition habits. Blood pressure medication increased urination at night, making it difficult to sleep. Participants on Coumadin had to avoid leafy green vegetables, which was described as a barrier to eating healthily.

Subcategory 2: informal caregivers. Family and friends facilitated healthy behaviors by providing encouragement and tangible support. Several participants were unable to cook or go grocery shopping, which meant they were reliant on others. Not surprisingly, being reliant on others for cooking often influenced dietary habits. Many informal caregivers recognized the importance of physical activity and adequate sleep and would often provide encouragement and tangible support (e.g., transportation or reminders).

Although informal caregivers provided support to engage in healthy behaviors, they also created circumstances that made it more difficult to engage in healthy behaviors. Some participants were more inactive because of an overprotective spouse or child who would limit activities and/or daily chores due to safety concerns. Participant #13 said, "Well, my kids, they always seem a little surprised at whatever I do. They think I'm supposed to be at home, I think, in a rocking chair so I don't hurt myself. They sometimes do too much for

me". Participants sometimes blamed sleep disturbances on family members who snored or called late at night.

Subcategory 3: seeking information. Most participants were active seekers of information to improve function and to determine why they had a stroke. Participants received health information from a wide variety of sources, such as watching television shows, searching websites, attending support groups, and reading books. Some participants noted that they mainly relied on formal or informal caregivers for health information, which was at times viewed as credible and helpful and was at other times viewed with skepticism and resentment. However, participants found it challenging to describe how they evaluated the credibility of health information. Some participants trusted most information when it came from an expert or someone they trusted (e.g., a doctor or friend). Other participants were skeptical of most health information, particularly nutrition information, because of contradictory reports or because they were worried about scams.

(3) Re-evaluation: priorities and attributions.

Subcategory 1: priorities and standards. Reduced autonomy in social roles, increased reliance on formal and informal caregivers, and a wealth of new information to process made participants reflect and re-evaluate priorities and self-image. Impairments made daily chores more challenging and time-consuming, which left less time to prioritize healthy behaviors. Many participants prioritized simple and familiar activities regardless of the known health consequences. In contrast, other participants described re-evaluating priorities to engage in multiple health behaviors.

Subcategory 2: stroke cause. The decision to prioritize single and multiple health behaviors was sometimes attributed to beliefs about what caused the stroke. For example, Participant #21 said,

"According to what I've read, the lack of exercise sometimes is a factor of diabetes, and it's also a factor of having a stroke. So if I want to avoid that pitfall in the future, it's incumbent upon me to keep myself in the best physical condition that I can, and that's what I'm trying to do [...] by exercising and eating healthy".

Alternatively, participant #1 said, "I don't think my diet has really changed because my stroke wasn't really due to my diet. So my diet has not changed [...], but I do try to engage in more activity".

Subcategory 3: self-image. Participants described a dynamic relationship between self-image and multiple healthy behaviors. Successfully engaging in healthy behaviors improved self-image and confidence. Achieving health-related goals provided a sense of accomplishment and control in maintaining independence. Some participants described having a more positive self-image because of increased spirituality and/or having a sense of purpose. Alternatively, participants' unsuccessful attempts at engaging in healthy behaviors decreased confidence and re-enforced a negative self-image.

(4) Resiliency: finding motivation and solutions.

Subcategory 1: self-determination. Participants routinely engaging in multiple health behaviors described having perseverance and persistence. Several participants described having an obstinate approach and refusing to have anything interfere with engagement. Participants successfully engaging in multiple health behaviors often had a "just do it"

attitude and were confident that nothing could get in their way. Participants determined to engage in multiple health behaviors frequently described it as a strategy to cope with stress and impairments.

Subcategory 2: strategic environmental planning. Participants described making their home environment conducive to multiple health behaviors. Participant #4 said,

"I just like to make sure that there's fruit and vegetables in the house and that I have access to them. And I –I've been trying to eat those rather than eating junk food, which I've been trying not to keep in the house".

Other participants described leaving exercise equipment in visible places and making their bedroom environment more conducive to sleep. One participant described not wanting their bedroom on the first floor because they wanted to stay physically active by having to climb stairs.

Subcategory 3: outcome expectations. Participants motivated to engage in multiple health behaviors were optimistic that maintaining engagement in multiple health behaviors would improve health and function. When asked why he was motivated to engage in physical activity, participant #5 said, "I want to get better. I want to be able to help myself more, you know. I just don't want to be dependent on everybody all the time". Participant #21 said, "Getting enough sleep and exercising are the things that contribute to me being able to mentally be stronger and help resist stress or negativity".

(5) Negative affectivity: anxiety and self-consciousness.

Subcategory 1: stress and anxiety. Impairments and an unknown trajectory of recovery were described as provoking stress, which in turn exacerbated symptoms, such as fatigue and pain. Some participants worried about another stroke, while others worried about becoming a burden on their families. Such worries resulted in chronic stress or anxiety that could prevent adequate sleep and subsequent engagement in physical activity and nutrition-related behaviors. Several participants stated that increased stress and/or inadequate sleep exacerbated symptom severity and resulted in having "good days and bad days", which influenced multiple health behaviors.

Subcategory 2: self-consciousness. Body-related, self-conscious emotions, such as shame and embarrassment, were described as influencing healthy behaviors in social and community settings. Some participants described being uncomfortable exercising at community centers. Participants also described limiting social and leisure physical activities because of difficulties walking and speaking. Participants with upper-extremity mobility impairments described reluctantly ordering unhealthy fried finger foods at restaurants to avoid the embarrassment and difficulties associated with eating healthier foods that required a fork, such as a salad. Body-related, self-conscious emotions and limiting options also propagated a negative self-image. Participant #12 said,

"It's just this and the effect the stroke had on me, and it's made me very sensitive (about my body). You know, I find myself questioning myself on everything I do—whether I'm doing the right thing or whether I'm doing it well enough".

**Table 5  Integration of qualitative and quantitative data.**

| Approach | Meta-inference | Quantitative data | Qualitative data |
|---|---|---|---|
| Behavior change principles | Reciprocal determinism correspondence | Moderate correlations between activities of daily living and multiple health behaviors; i.e., BMI ($r = -0.49$), physical activity (0.41), Sleep ($r = -0.55$), healthy eating habits ($r = 0.42$) | Descriptions of why restrictions in activities of daily living influence multiple behaviors and differences in how impairments influence single or multiple behaviors |
| | | Moderate correlation between hand impairments and healthy eating habits ($r = 0.45$) and non-significant correlations between hand impairments and physical activity ($r = 0.20$), sleep ($r = 0.20$), and BMI ($r = 0.05$) | Descriptions of how mobility impairments interact with environmental factors to restrict physical activity options only or restrict physical activity options and healthy eating habits |
| Global health/behavioral category | Circumstantial linkages | Small non-significant correlations between physical activity and nutrition ($r = 0.33$) and between physical activity and mobility ($r = 0.39$) | Descriptions of the linkages between physical activity and nutrition based on the circumstances of the participants |
| | | | Perceptions of how behaviors are prioritized and how traits may influence behaviors, such as how resiliency facilitates multiple behaviors and how negative affectivity hinders multiple behaviors |
| Association between behaviors | Sleep disturbance ripple effect | Moderate correlations between sleep and multiple health behaviors; BMI ($r = 0.48$), physical activity ($r = -0.48$), and ADL limitations ($r = -0.55$) | Descriptions of how sleep disturbances decreased motivation to engage in multiple health behaviors |

## Integrated results: meta-inferences and the conceptual model

We identified three meta-inferences: reciprocal determinism correspondence, circumstantial linkages, and the sleep disturbance ripple effect. Table 5 summarizes the data used to identify the meta-inferences. The conceptual model shown in Fig. 1 elaborates on the relationship between the thematic categories and quantitative data, which indicated factors that promote or hinder the engagement in multiple health behaviors.

## Meta-inference #1: reciprocal determinism correspondence
### *Behavioral change principles approach*

Reciprocal determinism refers to *Bandura*'s *(1986)* principle of how participants described dynamic interactions between person, behavior, and the environment. Correspondence refers to participants providing examples of reciprocal deterministic relationships that were relevant to multiple health behaviors and specific reciprocal deterministic relationships that were relevant to a single health behavior. As illustrated in Fig. 1, personal factors, such as limitations in accomplishing daily activities, interacting with environmental factors, such as inadequate social support, prompted changes in priorities and standards, which often resulted in frustration and stress that decreased motivation, confidence, and/or the ability to engage in multiple health behaviors. Alternatively, some participants described that a specific impairment, such as fatigue, would interact with a specific environmental factor, such as the inaccessibility of a recreational facility, to influence a single health behavior, such as physical activity. Correspondence was also supported by quantitative results. For example, activities of daily living—an indicator of multiple impairments or a more severe impairment—were inversely associated with multiple health behaviors, whereas problems with hand function—a specific impairment—were inversely associated with a single health behavior; i.e., nutrition.

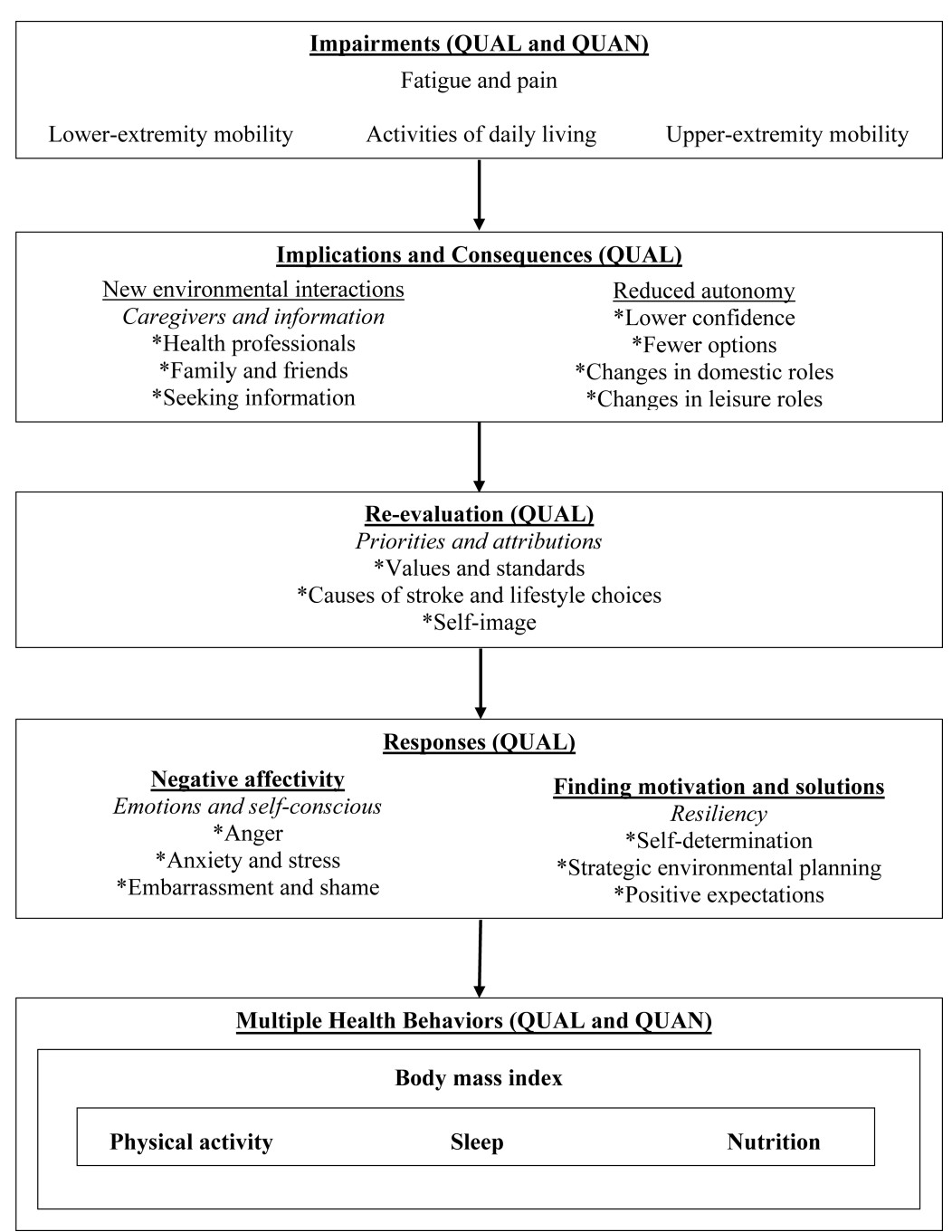

**Figure 1** **Conceptual model for engaging in multiple health behaviors among individuals with stroke.**
Note: QUAL, derived from qualitative data; QUAN, derived from quantitative data.

## Meta-inference #2: circumstantial linkages
### Global health/behavioral category approach

The term circumstantial linkages refers to relationships between behaviors and whether a factor, such as a global cognition or a psychological disposition, such as resilience or negative affectivity, influenced single or multiple behaviors depending on the circumstances
(e.g., environment, personality traits, and cognitions) of each individual. Although the participants described many scenarios that indicated possible links between global cognitions, behavioral categories, and specific behaviors (as suggested by *Noar, Chabot & Zimmerman, 2008*), these descriptions were not always consistent. For example, in the qualitative interviews, fatigue and pain were sometimes described as hindering single or multiple health behaviors, depending on environmental circumstances, such as the availability of tangible social support. Some participants described prioritizing one healthy behavior over another and rationalizing that one healthy behavior was more important than another healthy behavior. Other participants noted the importance of prioritizing multiple health behaviors, expressed confidence that the engagement in multiple health behaviors was completely under their control, and described personal traits, such as resiliency, that facilitated multiple behaviors.

Circumstantial linkages were also supported by the quantitative data, specifically the low-to-moderate correlations between different behaviors. For example, we found a small, non-significant correlation between physical activity and nutrition. Nonetheless, some participants described the importance of routinely engaging in both physical activity and eating healthily, whereas other participants described prioritizing either physical activity or eating healthily. If the participants were motivated to engage in both physical activity and eating healthily regardless of individual circumstances, it is likely that higher correlations would have been found.

### Meta-inference 3: sleep-disturbance ripple effect
*Multiple behavioral approach*

The sleep disturbance ripple effect refers to the influence of inadequate sleep on the factors that hinder the engagement in multiple behaviors. We found sleep disturbances to have moderate correlations with physical activity, BMI, and the ability to perform activities of daily living. In the qualitative data, sleep disturbances were described as increasing stress and fatigue, exacerbating impairments (e.g., pain, mobility, and fatigue), and decreasing the motivation to engage in multiple health behaviors, such as avoiding unhealthy temptations, cooking a healthy meal, and engaging in physical activity. Thus, both the qualitative and the quantitative data supported the importance of obtaining adequate sleep for engaging in multiple health behaviors.

## DISCUSSION

The science of how to promote multiple health behaviors is in its formative stage, which has considerable consequences for individuals with stroke. Targeting multiple behaviors has an 80% cumulative risk reduction in preventing secondary strokes (*Hackam & Spence, 2007*). Thus, there is a need to promote multiple health behaviors among individuals with stroke. We have conducted a novel mixed methods study to elaborate on the relationships between multiple health behaviors and to identify factors that facilitate and/or hinder multiple health behaviors. Our mixed methods results advance existing research by identifying meta-inferences that describe how stroke impairments may be barriers to single or multiple

health behaviors. We also developed a conceptual model depicting possible mediators that influence the relationship between stroke impairments and multiple health behaviors.

## Addressing purpose 1: stroke impairments and multiple health behaviors

Several participants described engaging in multiple health behaviors as a strategy to cope with stroke impairments. Quantitative studies of single health behaviors among individuals with stroke have also documented that emotional coping strategies and mood (*Visser et al., 2014*), physical activity (*Chen & Rimmer, 2011*), sleep (*Cavalcanti, Campos & Araüjo, 2013*), nutrition (*Serra et al., 2014*), and environmental factors (*Alguren et al., 2012*) influenced the process of adjusting to stroke impairments. Similarly, in a systematic review of qualitative studies (*Sarre et al., 2014*), resiliency, informal and formal caregivers, and engaging in healthy behaviors were described as influencing adjustment after a stroke. Thus, promoting engagement in multiple health behaviors may be a strategy to help individuals with stroke cope with their impairments and improve their quality of life.

However, the challenge in promoting multiple health behaviors is addressing the reciprocal relationship; i.e., impairments that make it difficult to engage in healthy behaviors. Future research should determine whether the negative impact of impairments on multiple health behaviors may be meditated by environmental factors (e.g., caregivers) and participants' characteristics (e.g., resiliency, mood, self-image, optimism, and perceptions about the causes of the stroke). Positive psychology constructs, such as resilience and optimism, are considered to be modifiable characteristics that may help facilitate the adjustment to a chronic disabling condition and the engagement in multiple health behaviors (*Martz & Livneh, 2015*).

## Addressing purpose 2: linkages among healthy behaviors

The results of this study showed that the linkages between healthy behaviors might be dependent on the characteristics of the participant. This finding is consistent with a recent research study of participants who were undergoing a rehabilitation program. The results of that quantitative study indicated that the relationship between physical activity and nutrition was mediated by habit strength and transfer cognition (*Fleig et al., 2014*). Research has also indicated that promoting the engagement in one behavior (e.g., physical activity) might promote the engagement in another behavior (e.g., reducing cigarette craving) (*Haasova et al., 2013*; *Noar, Chabot & Zimmerman, 2008*). Mixed methods studies of multiple health behaviors have mainly focused on weight management among non-disabled children and adults. Such studies have documented circumstances in which people can increase physical activity levels and/or develop healthy eating habits (*Abildso et al., 2010*; *James et al., 2016*; *Sliwa et al., 2014*).

Consistent with the existing research, we found that sleep had moderate correlations with physical activity, BMI, and the ability to perform activities in daily living (*Bakken et al., 2012*; *Cavalcanti, Campos & Araüjo, 2013*; *Grandner et al., 2011*). The participants described that sleep disturbances resulted in fatigue and negative emotions, which decreased their motivation to engage in multiple health behaviors. A mixed methods study of sleep habits among

patients with traumatic brain injury showed that inadequate sleep had a major impact on health outcomes and reduced adherence to rehabilitation activities (*De La Rue-Evans, Nesbitt & Oka, 2013*). Future studies of individuals with stroke should examine whether sleep disturbances need to be addressed before multiple behavior changes can occur.

## Addressing purpose 3: developing multiple behavior change interventions

The results indicated that future research should examine the efficacy of multiple behavioral change interventions that focus on increasing resiliency and self-confidence and that address negative emotions and sleep disturbances. Such multiple behavioral change interventions might include education about identifying creditable sources of information, re-arranging the social and physical environment, promoting social support, exploring participants' perceptions about what caused their stroke, and providing examples of how engaging in multiple health behaviors may help reduce the need for medication. Multiple behavioral change interventions may also need to target participants who rationalize the prioritization of one particular healthy behavior over another.

In a recent review of qualitative studies, *Lawrence et al. (2016)* reported that individuals with stroke generally perceived that participating in behavior change interventions was beneficial. They described feeling supported and that they acquired new knowledge and gained increased confidence to engage in healthy behaviors. These findings were consistent with our recommendation to identify creditable sources of information and increase confidence and social support in the design of multiple behavioral change interventions. Therefore, existing interventions could be adapted to promote multiple behavioral changes in individuals with stroke.

Self-management interventions focus on increasing confidence and teaching skills, such as problem-solving and resource utilization, which may be relevant to promoting engagement in single or multiple behaviors to manage single or multiple impairments (*Lo et al., 2013*; *Lorig & Holman, 2003*; *Parke et al., 2015*). However, the efficacy and theoretical underpinnings of these interventions to support multiple behavioral changes are unclear. The results of a mixed methods study of a tailored self-management program showed promise in promoting patient activation, exercise, and healthy eating (*Montgomery et al., 2015*). Future studies should incorporate objective and self-report measures of healthy behaviors to determine whether self-management interventions can promote multiple health behaviors.

### Application to *Noar, Chabot & Zimmerman's (2008)* framework

We identified three meta-inferences that were consistent with *Noar, Chabot & Zimmerman*'s *(2008)* three approaches to understanding and promoting multiple behavioral changes. Regarding the use of the behavioral change principle approach, promoting multiple health behaviors simultaneously may require identifying and addressing reciprocal deterministic relationships that correspond with multiple health behaviors. Similarly, identifying and addressing the relationships that correspond with a single health behavior may be a strategy to promote multiple behavior changes sequentially. Regarding the global health/behavioral category approach, targeting and improving global cognitions and

dispositions, such as resiliency and optimism, may help promote multiple health behaviors. However, further research is needed to determine the circumstances in which global cognition could be addressed. Although targeting global cognitions has the allure of promoting multiple behaviors simultaneously, they may be harder to change than cognitions about particular behaviors (*Magidson et al., 2014*). Furthermore, successfully changing global cognitions may not always influence behaviors in the same way in all participants (*Blackie et al., 2014*). Regarding the multiple behavioral approach, we found that it might be important to target sleep disturbances to promote multiple behavior changes.

### Limitations

The limitations of this study include the small sample size, the cross-sectional design, and the generalizability of the results. Because little is known about the linkages between multiple health behaviors, which makes it difficult to generate *a-priori* hypotheses, we decided to prioritize qualitative data over quantitative data. Therefore, although the sample size was sufficient for the qualitative analysis, it limited the findings of the quantitative analysis. The small sample size precluded the ability to conduct a valid multiple regression analysis, while the cross-sectional design precluded the ability able to make inferences about causal relationships. Furthermore, the measures of physical activity and nutrition were not specifically validated in individuals with stroke, and they may have influenced correlations in unknown ways. The generalizability of the study is limited because of the nature of the qualitative analysis and the small sample size used in the quantitative analysis. We did not include adults with severe aphasia or cognitive impairments. Thus, results may not be relevant to individuals with stroke with these severe impairments. In addition, the recruitment of the first focus group was potentially biased because it was limited to the participants in a physical therapy group exercise class. Finally, the incorporation of the additional measures of personality and cognition could have helped promote greater triangulation of the findings. A sequential mixed design that analyzed qualitative data first would have enabled the selection of additional questionnaires to facilitate triangulation.

## CONCLUSION

Coping skills, social support, and type and extent of impairments are just some of the complexities that may influence engagement in multiple health behaviors among adults with disabling conditions. The use of mixed methods can help examine such complexities from different angles to offer a more holistic prospective of how to promote engagement in multiple health behaviors. Researchers in rehabilitation science and public health have noted the need to use mixed methods to understand the complex biopsychosocial interactions that result in participation restrictions for engaging in healthy behaviors (*Brownson, Roux & Swartz, 2014*; *Kroll, 2011*).

Regarding the question about whether multiple behavior change interventions should target behaviors sequentially and/or simultaneously, the answer may depend on the characteristics of the participants. We found examples of facilitators and barriers that were described as being relevant to a single behavior and in other circumstances as being relevant to multiple health behaviors. Therefore, the extent to which a particular facilitator and/or

barrier influences multiple behaviors may depend on cognition and environmental circumstances. Examining the circumstances in which factors influence single or multiple behaviors and understanding when it is possible to change factors that promote multiple behavior changes will be vital in developing multiple behavior change interventions among individuals with stroke. Further mixed methods research is needed to identify the circumstances that influence single or multiple behaviors and the particular factors that may influence multiple behaviors regardless of the circumstances.

### Funding
The research is funded by the American Heart Association (Grant #11BGIA7710003) and supported by NIH StrokeNet (Cleveland Regional Coordinating Center). The funders had no role in study design, data collection and analysis, decision to publish, or preparation of the manuscript.

### Grant Disclosures
The following grant information was disclosed by the authors:
American Heart Association: #11BGIA7710003.
NIH StrokeNet (Cleveland Regional Coordinating Center).

### Competing Interests
The authors declare there are no competing interests.

### Author Contributions
- Matthew Plow conceived and designed the experiments, performed the experiments, analyzed the data, wrote the paper, prepared figures and/or tables, reviewed drafts of the paper.
- Shirley M. Moore, Martha Sajatovic and Irene Katzan analyzed the data, wrote the paper, reviewed drafts of the paper.

### Human Ethics
The following information was supplied relating to ethical approvals (i.e., approving body and any reference numbers):
  University Hospital granted approval to carry out the study (IRB Application Ref: 11-847).

### Data Availability
  The raw data has been supplied as Data S1.

### Supplemental Information
Supplemental information for this article can be found online at http://dx.doi.org/10.7717/peerj.3210#supplemental-information.

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
