# Peer review of "A mixed methods study of multiple health behaviors among individuals with stroke"

_PeerJ, doi:10.7717/peerj.3210_

## Round 0.1 · original submission · Minor Revisions

· Academic Editor

Minor Revisions

I have personally reviewed the MS and concur with the issues raised by the reviewers.

Reviewer 1 ·

Basic reporting

No comment

Experimental design

No comment

Validity of the findings

No comment

Additional comments

-Very well written manuscript with a difficult task of integrating qualitative analysis with quantitative data from individuals with stroke.

General Comments:
-The use of stroke survivors is not patient first and heavily repeated in the entire article. It is very repetitive in nature, please address.
-Line 76- The statement needs to be expanded to further elucidate secondary prevention interventions.
-Line 80- please make a statement and not a question.
Methods:
- Please clarify did you only evaluate hand usage in the effected limb and was this the subject's dominant hand.
-line 157- please provide justification why you multiplied the activities by 9, 5, 3 and not use raw numbers?
-Please provide clarification if all the data was collected in the same order with each subject?
-line 304- This is the first use of the words pilot study, please indicate this earlier in the manuscript.
-Line 304- remind the reader again of your first purpose again.
-Line 316-319- Statement very vague, please expand here to add clarity to your point.
-Table 4 was listed to be inserted twice
-Line 330- might be beneficial to reader to add clarity on "information" like to seeking information or lack of information.
Line 332-334: To add further clarity, please remind the readers in paraphrase on the aims of your study and the relation to the categories.
Line 595-597: Sentence needs to be broken into two sentences and expanded upon to provide further clarity.
Line 652: Remove reference to introduction and paraphrase your aim. Helpful to remind the reader again.
Table 1: Cane use, walker use , wheelchair use: Please add these numbers again, they do not equal 25.
Table 2 or text of methods: indicate impaired limb tested and do you have dominance hand data, if so add to table 1.

·

Basic reporting

Very clearly written and well-constructed paper. Good reference to the wider relevant literature. Context clearly described and, the argument and need for the study well and clearly articulated.

Experimental design

Well defined and justified research questions and design. The authors appear to have been rigorous throughout with regards to the level of detail provided as well as all aspects of their decision making, ensuring quality and rigour, and providing an audit trail of this.

Validity of the findings

Findings are novel and make a valuable contribution to the evidence base on stroke secondary prevention and self-management. All examples given appear to be well grounded in the data and interpretations of this appear meaningful and robust.

Additional comments

I thought this was an excellent paper. It was very well constructed and easy to follow and sufficient justification provided for each step of decision making along the way. I think this paper and the findings from this work will make a very important contribution to the evidence base. My only comments would be:

1. that it wasn't entirely clear whether the focus group and telephone interview participants were the same people, and the numbers of telephone interviews conducted - I presume these were in fact member checks but perhaps this detail could be added for clarity since it is stated that some areas were further developed and probed on within the telephone interviews.
2. I wonder whether the subcategories could be labelled 'subthemes' for consistency of reporting with the main themes in the qual analysis.
3. The limitations of the work are acknowledged but it seems to me that something needs to be added around the fact that participants were those who clearly did not suffer from a significant degree of communicative impairments and I wonder whether something needs to be said about the transferability of the findings to stroke survivors with significant aphasia and other types of impairment that do not seem to have been characteristic of the sample in the study.

---

## Round 0.2 · accepted · Accept

· Academic Editor

Accept

Thank you for your responses to the reviews, Congratulations. Good Job!